# DeepGEM: Generalized Expectation-Maximization for Blind Inversion

**Angela F. Gao**[1]     **Jorge C. Castellanos**[2]

**Yisong Yue**[1]     **Zachary E. Ross**[2]     **Katherine L. Bouman**[1]

[1]Computing and Mathematical Sciences, California Institute of Technology
[2]Caltech Seismological Laboratory, California Institute of Technology
`{afgao, jcastellanos, yyue, zross, klbouman} @ caltech.edu`

## Abstract

Typically, inversion algorithms assume that a forward model, which relates a source to its resulting measurements, is known and fixed. Using collected indirect measurements and the forward model, the goal becomes to recover the source. When the forward model is unknown, or imperfect, artifacts due to model mismatch occur in the recovery of the source. In this paper, we study the problem of blind inversion: solving an inverse problem with unknown or imperfect knowledge of the forward model parameters. We propose DeepGEM, a variational Expectation-Maximization (EM) framework that can be used to solve for the unknown parameters of the forward model in an unsupervised manner. DeepGEM makes use of a normalizing flow generative network to efficiently capture complex posterior distributions, which leads to more accurate evaluation of the source's posterior distribution used in EM. We showcase the effectiveness of our DeepGEM approach by achieving strong performance on the challenging problem of blind seismic tomography, where we significantly outperform the standard method used in seismology. We also demonstrate the generality of DeepGEM by applying it to a simple case of blind deconvolution.

## 1 Introduction

Physics-based inversion methods typically recover an unknown source from indirect measurements by assuming that the source and measurements are related via a known forward model [15, 7, 30]. For example, non-blind deconvolution algorithms often assume that a measured blurry image is related to its true sharp image via a known spatially-invariant blur kernel [9]; and traditional seismic inversion methods assume that the spatially-varying velocity of the Earth's interior is known *a priori* when solving for an earthquake's hypocenter [30]. However, these "known" forward models are generally idealized and ignore intricacies of the systems that are either hard to capture or simply unknown. Inversion algorithms with forward *model mismatch* result in biased reconstructions. For instance, bias is regularly seen in non-blind deconvolution results, where reconstruction artifacts are often present due to the use of an incorrect blur kernel [9].

In order to reduce the effects of model mismatch, in this paper we tackle the problem of *blind inversion*: solving an inverse problem without knowledge of the underlying forward model parameters. When considering how learning can help, a natural first idea might be to learn a direct map from measurements to the desired source via supervised learning. However, such a "model-free" approach is generally not practical, due to the lack of available ground truth training data. For example, the blur kernel caused by handheld camera shake cannot be reproduced to get a training set of

35th Conference on Neural Information Processing Systems (NeurIPS 2021), .

sharp-blurry pairs to train a deconvolution approach. In seismic tomography, synthetic earthquakes cannot be placed densely throughout the interior of the Earth to measure the ground response as a function of hypocenter location. An alternative approach, which we adopt, is to develop unsupervised methods that treat the true forward model as something that is unobserved and must be inferred. An additional consideration is that we must solve for the true forward model from a single dataset, without knowledge of the true source that produces the measurements

In this paper, we propose Deep Generalized Expectation-Maximization (DeepGEM) for solving blind inverse problems. Using the indirect measurements as input, DeepGEM jointly estimates the source and forward model that together produce the observed measurements. DeepGEM is a variational inference based framework that makes use of deep learning machinery to easily capture and optimize complex probabilistic distributions that cannot be easily integrated in analytic Expectation-Maximization (EM) solutions. Our proposed framework is generic and can be applied to blind inversion problems described by differentiable forward models. In Section 4 we showcase the effectiveness of our DeepGEM approach on the challenging problem of blind seismic tomography, where we significantly outperform methods used in seismology. We also demonstrate the generality of DeepGEM by applying it to blind deconvolution in Section 5.

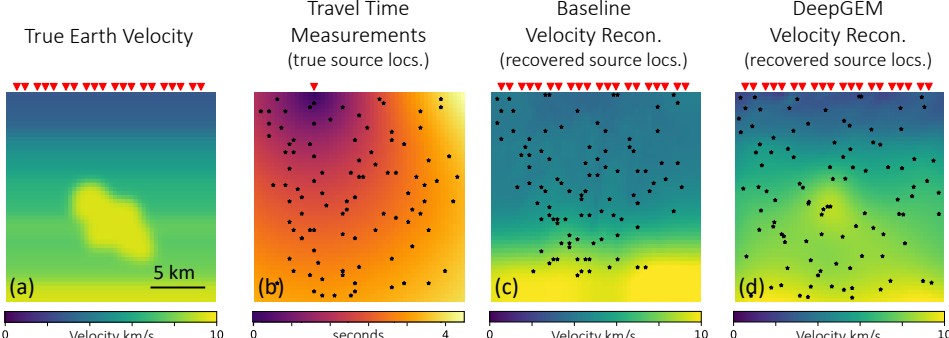

Figure 1: **DeepGEM applied to blind seismic tomography.** (a) A simulated cross section of the Earth's interior (velocity structure), along with the locations of receivers on the surface (red triangles) that collect measurements. (b) The time it takes for a wave traveling from a source below the surface to reach the specified receiver is visualized for each location in the region of interest. The overlaid dots represent the true locations of simulated earthquakes and indicate the measured travel times that constrain optimization. (c) The subsurface velocity reconstruction obtained using a baseline approach optimized with the help of a seismologist. Note that the bright anomaly is missing from this reconstruction. Overlaid dots represent the inferred earthquake locations. (d) DeepGEM reconstructed subsurface velocity and inferred earthquake locations. Note that DeepGEM is able to accurately recover the gradient of the velocity field as well as partially recover the central anomaly.

## 2    Background and related work

The joint optimization of a forward model with source recovery is a very challenging ill-posed problem, leading to many possible solutions that are hard to disambiguate. For example, in blind deconvolution the blurry image observed can be equivalently explained by a sharp source image convolved with an extended kernel or a blurry source image convolved with a impulse kernel; to prefer one solution over another, additional information, such as image priors, must be considered. Previous work on inversion in poorly characterized systems focused on limited contexts, such as spatially-invariant blind deconvolution [21, 22, 14, 10] and CT with a simple rotational error [35]. These methods tend to be highly specialized to each application domain, and cannot be easily generalized. In contrast, we propose a flexible model-based Bayesian framework that can be applied across multiple differentiable blind inversion problems.

### 2.1   Model-based Bayesian inversion

In model-based inversion, unobserved sources $x$ and observed measurements $y$ are related through a forward model: $y = f(x)$. When the parameters of the underlying model are unknown, one can parameterize the forward model as $f_\theta(x)$ and then solve for the true model parameters $\theta^*$. A common approach is to solve a *maximum a posteriori* (MAP) objective: either $\text{MAP}_{\theta,x}$ or $\text{MAP}_\theta$.

$\text{MAP}_{\theta,x}$ solves for the optimal point estimate of the pair $\{\hat{\theta}, \hat{x}\}$ that maximizes a joint objective:

$$\{\hat{\theta}, \hat{x}\} = \arg\max_{\theta,x} [\log p(\theta, x|y)] = \arg\max_{\theta,x} [\log p(y|\theta, x) + \log p(\theta) + \log p(x)]. \quad (1)$$

In practice, formal probabilistic definitions of $p(\theta)$ and $p(x)$ are often unknown and replaced with regularization terms (e.g., total variation, sparsity) [22, 21]. Although $\text{MAP}_{\theta,x}$ provides a straightforward approach to solve for $\theta$, the energy landscape of Eq. 1 is typically poorly behaved due to the ill-posed nature of the problem [22]. As a result, optimization is likely to get stuck in a (bad) local minimum.

$\text{MAP}_{\theta}$ attempts to smooth the energy landscape by reducing the number of parameters that must be optimized. This is done by solving for parameters of the forward model, $\theta$, that perform best under the full volume of possible $x$ interpretations:

$$\hat{\theta} = \arg\max_{\theta} [\log p(\theta|y)] = \arg\max_{\theta} \left[ \log \left( \int_x p(y|\theta, x) p(x) dx \right) + \log p(\theta) \right]. \quad (2)$$

Since this marginalization integral is often intractable, Expectation-Maximization (EM) algorithms have long been used for solving $\text{MAP}_{\theta}$ efficiently [11]. EM is an iterative algorithm that alternates between: "E"-Step) calculating the posterior of $x$ conditioned on the current estimated forward model parameters $\theta^{(t-1)}$; and M-Step) updating $\theta$ to maximize the expected value of the log likelihood:

$$\theta^{(t)} = \arg\max_{\theta} \left[ \mathbb{E}_{x \sim p(x|y, \theta^{(t-1)})} [\log p(y|\theta, x)] + \log p(\theta) \right]. \quad (3)$$

The advantage of $\text{MAP}_{\theta}$ over $\text{MAP}_{\theta,x}$ optimization for the blind deconvolution problem was described in [22] and its success demonstrated via EM optimization in [21]. However, it is important to note that evaluating the expectation in of Eq. (3) over complex distributions is often intractable. For instance, the authors of [21] were forced to restrict the posterior distribution to a Gaussian distribution. Alternatively, stochastic EM methods [5, 8] bypasses the need to evaluate the expectation directly, approximating it by sampling the distribution. In this paper, we solve $\text{MAP}_{\theta}$ using complex distributions parameterized by deep neural networks.

**Forward model parameterization** Model-based inversion requires that the parametric form of $f_{\theta}(x)$ is well matched with the true forward model, which is not always known. Alternatively, neural networks can be used to approximate the forward model, where $\theta$ represents the network weights. This setup is flexible, in that it can be used to approximate arbitrarily complex forward models, with the downside that it is often not interpretable when the parameters have no physical meaning. In this work, we develop and make use of interpretable, physically-motivated deep neural networks to parameterize $f_{\theta}(\cdot)$ for the problems of blind seismic tomography and blind deconvolution.

## 2.2 Deep variational distributions

We are interested in solving blind inverse problems using the EM algorithm to optimize the $\text{MAP}_{\theta}$ objective. Optimizing in this fashion requires the use of the posterior distribution $p(x|y, \theta^{(t-1)})$ in evaluating Eq. 3. As inverse problems are often ill-posed, we expect that the posterior distribution of source $x$ conditioned on the forward model parameters $\theta$ is likely to be complex and perhaps even multi-modal. Therefore, it is important to be able to parameterize a flexible family of distributions to best estimate this conditional posterior.

Deep Probabilistic Imaging (DPI) [32] uses a normalizing flow-based generative model, $G_{\phi}(\cdot)$, to solve for the uncertainty of an unknown source $x$ given a fixed forward model $f(x)$ and measurements $y$. DPI solves the variational objective:

$$\hat{\phi} = \arg\min_{\phi} \text{KL}(q_{\phi}(x)||p(x|y)) = \arg\min_{\phi} E_{x \sim q_{\phi}(x)} [-\log p(y|x) - \log p(x) + \log q_{\phi}(x)], \quad (4)$$

where $q_{\phi}(x)$ is the implicit distribution defined by $G_{\phi}(z)$ for $z \in \mathcal{R}^{|x|} \sim \mathcal{N}(0,1)$[1], and $\log p(y|x) \propto ||y - f(x)||_2 + c$ when there exists *i.i.d.* Gaussian noise on the measurements, $y$. After inference, $q_{\phi}(x)$ can be efficiently sampled by evaluating $G_{\phi}(z)$ for a Gaussian sample $z$.

The DPI variational objective is equivalent to the Variational Autoencoder [17, 28] objective, except with a fixed decoder, $f(x)$. In practice, vanilla VAEs constrain the posterior to be a Gaussian

---

[1] $|x|$ is the dimension of $x$

distribution, relying on the reparameterization trick for tractable optimization. Alternatively, DPI uses flow-based networks to efficiently sample and directly compute $q_\phi(x)$ [12, 13, 18]. DPI's use of a flow-based network allows for complicated and multi-modal posterior distributions constrained only by the space of possible invertible network architectures. Our proposed DeepGEM approach utilizes similar tools to model flexible distributions over $x$, while simultaneously learning the forward model parameters $\theta$.

## 3    Methods

We propose a deep variational EM approach (DeepGEM) that optimizes the MAP$_\theta$ objective in Eq. 2 to recover the parameters of a forward model $f_\theta(x)$ using only a single static set of measurements $y$. Once learned, the updated forward model can then be used to estimate the posterior distribution of the unknown source, $x$. DeepGEM iterates between two stages that are inspired by the standard EM algorithm: (1) an "E"-step that learns a variational distribution, $q_{\phi^{(t)}}(x)$, to approximate the posterior distribution of $x$ given the current forward model parameters $\theta^{(t-1)}$, and (2) an M-step (refer to Eq. 3) that solves for $\theta^{(t)}$ that maximizes the expected value of the log likelihood function of $\theta$, with respect to the posterior distribution $q_{\phi^{(t)}}(x)$ estimated in the prior "E"-step. Each step is alternated and solved to convergence. Since this is an EM algorithm (up to the variational approximation), our method inherits properties of EM, including convergence to a local minimum of Eq. 2.

### 3.1    "Expectation" step ("E"-step)

The goal of DeepGEM's "E"-step is to solve for the posterior distribution $p(x|y, \theta^{(t-1)})$ that facilitates optimizing Eq. 3. Because this posterior distribution can be very complex, and even multi-modal, we propose to use a flexible variational approach to learn the parameters $\phi$ of an approximate posterior distribution $q_\phi(x)$. The variational distribution $q_\phi(x)$ can then be used to evaluate Eq. 3 via efficient sampling.

Using DPI we solve for a flexible variational distribution, $q_\phi(x)$ that well approximates the posterior distribution $p(x|y, \theta^{(t-1)})$. DPI uses a normalizing flow network, $G_\phi(z)$, with input $z \in \mathbb{R}^{|x|}$, where $x = G_\phi(z) \sim q_\phi(x)$ when $z \sim \mathcal{N}(0, \mathbb{1})$. Normalizing flow networks allow for exact computation of the log-likelihood $\log q_\phi(x)$ needed to solve

$$\phi^{(t)} = \arg\min_\phi \mathrm{KL}(q_\phi(x)||p(x|y, \theta^{(t-1)}))$$

$$\approx \arg\min_\phi \frac{1}{N} \sum_{n=1}^N [-\log p(y|\theta^{(t-1)}, x_n) - \log p(x_n) + \log q_\phi(x_n)]$$

$$\text{for} \quad x_n = G_\phi(z_n), \quad z_n \sim \mathcal{N}(0, \mathbf{1}), \tag{5}$$

(as derived from Eq. 4) for a batch size of $N$ where $\log p(x)$ is a prior on the source and $\log p(y|x, \theta^{(t)})$ is the data likelihood. When assuming the measurements $y$ experience *i.i.d* additive Gaussian noise with standard deviation $\sigma_y$, $\log p(y|\theta^{(t)}, x_n) = \frac{1}{2\sigma_y^2} \|y - f_{\theta^{(t)}}(x_n))\|^2 + c$.

### 3.2    Maximization step (M-step)

The goal of DeepGEM's M-step is to use the parameterized approximate posterior distribution, $q_{\phi^{(t)}}(x)$, from the prior "E"-step to update $\theta$, the parameters of the unknown forward model $f_\theta(\cdot)$. This is achieved by sampling from the learned normalizing flow network, $G_{\phi^{(t)}}(\cdot)$, to stochastically solve Eq. 3 :

$$\theta^{(t)} \approx \arg\max_\theta \left[ \frac{1}{N} \sum_{n=1}^N [\log p(y|\theta, x_n)] + \log p(\theta) \right] \quad \text{for} \quad x_n = G_{\phi^{(t)}}(z_n), \ z_n \sim \mathcal{N}(0, \mathbb{1}), \tag{6}$$

where $p(\theta)$ is a prior on the forward model. This prior can be used to encourage the forward model parameters to remain close to an initial model $\tilde{\theta}$ by defining $\log p(\theta) \propto ||\theta - \tilde{\theta}||_2 + c$.

# 4 Case study: blind seismic tomography

Two fundamental seismic inverse problems are spatially localizing an earthquake's hypocenter (also referred to as source localization) and tomographic reconstruction of the Earth's subsurface [30]. These problems are interconnected: source localization relies on knowing how fast waves propagate through different regions of the Earth's interior, referred to as the Earth's *velocity* structure. However, in standard seismological practice, due to difficulties in solving these problems jointly, they are generally treated separately: source localization is performed initially using oversimplified velocity models [30], and then the tomography problem is performed by taking those best-fitting hypocenters as ground truth [26]. This approach typically results in the need to over-regularize the inverse problem by smoothing out high-frequency information [2], and can only be improved by carefully incorporating other forms of information such as waveform-derived quantities [16, 23, 4, 24] or by performing costly experiments such as controlled explosions. In contrast, we demonstrate our DeepGEM approach on blind seismic tomography, solving for the subsurface velocity (parameterized by $\theta$) when the source hypocenters, $x$, are unknown. Measurements, $y$, used to constrain the inverse problem are the time it takes for the first wave to propagate from its source to a receiver on the Earth's surface, referred to as a *travel time* measurement (refer to Fig. 1).

## 4.1 Seismic tomography background

**Physics of earthquake source localization:** The earthquake source location, also called the hypocenter, is the location where the earthquake nucleates [30]. The source location can be triangulated using travel times from multiple receivers. However, when there are very few receivers ($< 3$ in 3D, $< 2$ in 2D), the source localization is ill-posed and there exists multiple equally optimal solutions.

**Physics of travel time tomography:** Travel time tomography is a method for reconstructing the velocity structure using the absolute arrival times of earthquake waves from the earthquake to the receiver [30]. Given perfect knowledge of the earthquake locations $x$ and receivers $r$ at every position in the ground, the exact velocity can be computed by solving the Eikonal equation $V(r) = \|\nabla_r T(x, r)\|_2^{-1}$, where $T(x, r)$ is the travel time from an earthquake to a receiver. There exists an analytical solution to the Eikonal equation; however, seismologists often use simplifications, such as straight ray tomography, for efficiency [2].

**Deep Learning for travel time tomography:** EikoNet [31] implicitly solves the Eikonal equation [33] by learning a mapping from a source-receiver pair $(x, r)$ to its associated travel time, learning $\theta$ such that $f_\theta(x, r) \approx T(x, r)$ where $T(x, r)$ is the true travel time. The velocity structure can then be extracted from the learned EikoNet by solving the Eikonal equation through automatic differentiation. In particular, the computed velocity $V(s)$ at location $s$ is $\|\nabla_s f_\theta(x, s)\|_2^{-1}$. Note that this computed velocity depends on the source location $x$. Ideally, the velocity should be invariant to the source location; however, in practice, this is only true when EikoNet is trained with densely-sampled $(x, r)$ pairs. Additionally, the travel time from a source to a receiver, $T(x, r)$, should be identical to the travel time from a receiver to a source, $T(r, x)$, but remains unconstrained by EikoNet.

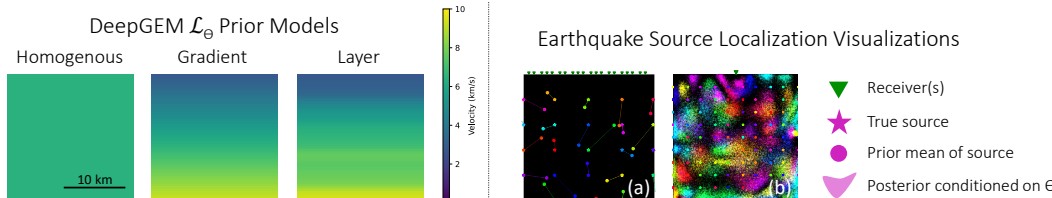

Figure 2: (Left) A visualization of the homogeneous, gradient, and layer models used in $p(\theta)$'s $\mathcal{L}_\theta$ term. (Right) Visualizations used to describe source configurations and earthquake source posteriors. (a) Visualizations of the true and initialized source locations are plotted as stars (true) connected to circles, which indicate the expected source locations according to $p(x)$. Note that the expected source locations deviate significantly from the true locations. (b) Visualizations of the learned posterior distribution, $q_\phi(x|y, \theta)$, for each source are plotted as colored histograms and overlaid with stars (of the same color) indicating the true source location.

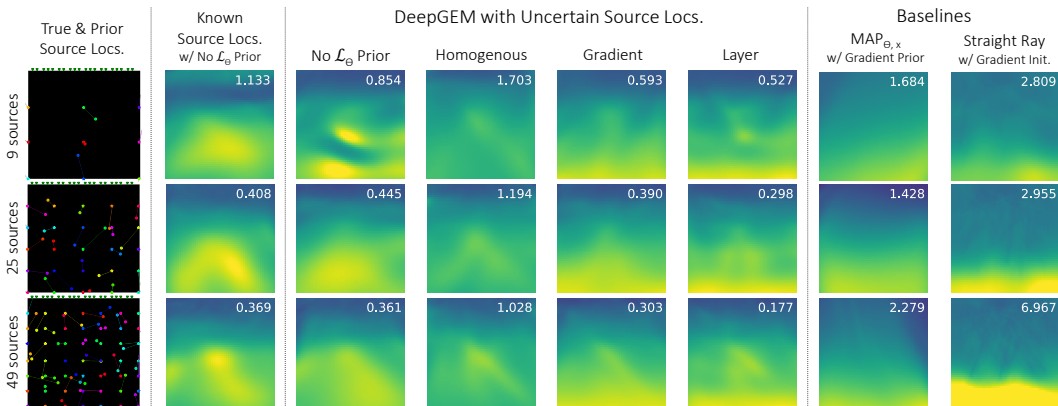

Figure 3: **DeepGEM reconstructions significantly outperform baselines, and improve with more sources and better $\mathcal{L}_\theta$ prior models.** Each row corresponds to the reconstructions obtained from a single noisy observation of travel times, where the true Earth velocity is shown in Fig. 1(a). Results shown are simulated using 20 surface receivers and a varying number of sources (9, 25, and 49) in a uniform grid. Columns 3-6 show DeepGEM results obtained using different $\mathcal{L}_\theta$ priors. Note that results improve as the $\mathcal{L}_\theta$ prior becomes closer to the true velocity structure, and as the number of sources increases. As a reference, column 2 shows the velocity reconstruction obtained using DeepGEM under fixed, perfectly known source locations. Columns 7-8 show results obtained by the baseline approaches. The velocity reconstruction MSE is included in the top right of each reconstruction. DeepGEM substantially outperforms both straight ray and MAP$_{\theta,x}$ baselines. Note that DeepGEM also sometimes outperforms the second column, where the source locations are known; this is due to the fact that DeepGEM is sometimes able to absorb measurement error due to its overparameterization. See the supplemental material for more examples.

## 4.2 DeepGEM setup for blind seismic tomography

For blind seismic tomography, we parameterize the forward model using a modification of EikoNet, $f_\theta(x, r)$, with unknown source location $x$ and known receiver location $r$ as inputs and $y \approx T(x, r)$, the absolute travel time between $x$ and $r$, as output. In order to solve Eqs. 5 and 6 for an updated forward model, we must define priors $p(x)$ and $p(\theta)$. The prior over source locations, $p(x)$, is often well defined, typically a Gaussian distribution $\mathcal{N}(\bar{x}, \sigma_x)$ with a standard deviation of $\sigma_x \sim 2$ km and mean $\bar{x}$.

We construct a prior over the forward model, $p(\theta)$, that encourages EikoNet to learn a velocity close to $\tilde{V}(s)$. Additionally, as discussed above, there are constraints specific to seismic tomography that are not explicitly enforced through EikoNet's architecture: (1) velocity reconstruction invariance with respect to the source location, and (2) travel time symmetry between sources and receivers. We augment the prior $p(\theta)$ to include these constraints, implemented as soft constraints:

$$\log p(\theta) = \lambda_\theta \overbrace{\sum_{\substack{r \in \mathcal{R}, \\ s \in \mathcal{S}}} \|\tilde{V}(s) - V_r(s)\|_2}^{\mathcal{L}_\theta} + \lambda_V \overbrace{\sum_{\substack{r_i, r_j \in \mathcal{R}, \\ s \in \mathcal{S}}} \|V_{r_i}(s) - V_{r_j}(s)\|_2}^{\mathcal{L}_V} + \lambda_T \overbrace{\sum_{\substack{r \in \mathcal{R}, \\ s \in \mathcal{S}}} \|T(s, r) - T(r, s)\|_2}^{\mathcal{L}_T} .$$
(7)

The velocity constraint is represented through $\mathcal{L}_V$ and travel time constraint through $\mathcal{L}_T$, with corresponding hyperparameters $\lambda_V$ and $\lambda_T$. $\mathcal{S}$ is a set of points sampled uniformly from the region of interest, $\mathcal{R}$ is the set of all receiver locations, and $V_r(s) = \|\nabla_s f_\theta(r, s)\|_2^{-1}$.

**Implementation details:** In our experiments we define a realistic prior over unknown source locations as $p(x) = \mathcal{N}(\bar{x}, \sigma_x)$, where $\bar{x} \sim \mathcal{N}(x, \sigma_x)$ and $\sigma_x = 2$ km. We assume the measurements $y$ are sampled from a Gaussian distribution with mean $\tilde{y}$ – true travel times computed using the package `eikonalfm` [29, 33] – and a realistic standard deviation of $\sigma_y = 0.2$ seconds. To simulate real world passive tomography, we assume receivers are located only along the surface (the top edge of the 2D image) of the region of interest, which is 20 km $\times$20 km, and sources can be anywhere within the region of interest. Before DeepGEM optimization, we first initialize $\theta$ using maximum likelihood estimation assuming $\bar{x}$ is the true source location.

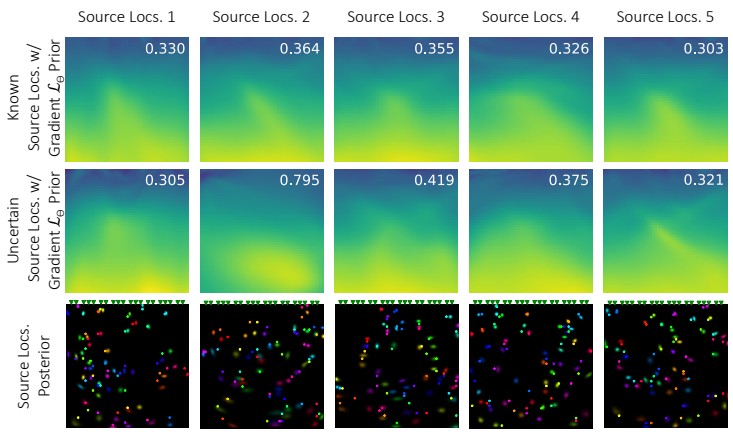

Table 1: Velocity and source localization error on random source configurations

| Source Count | Velocity Error (km/s) | Source Loc. Error (m) |
|---|---|---|
| 9 | 0.52 ± 0.063 | 62.88 ± 21.77 |
| 25 | 0.42 ± 0.043 | 36.62 ± 4.93 |
| 49 | 0.44 ± 0.181 | 33.78 ± 4.56 |
| 100 | 0.27 ± 0.021 | 30.38 ± 1.43 |

Figure 4: **DeepGEM consistently recovers prominent features across various earthquake source configurations.** Row 2 shows DeepGEM results obtained using different random source configurations, where the true Earth velocity is shown in Fig. 1(a). As a reference, row 1 shows the velocity reconstruction obtained using DeepGEM under fixed, perfectly known source locations. As can be seen by the table, both velocity and localization error decrease with an increasing number of sources in the region of interest. Each mean and standard deviation is computed using 5 random realizations of true source configurations.

The posterior distribution of the source locations, $x$, is estimated using a Real-NVP network $G_\phi(\cdot)$ with 16 affine coupling layers. An updated EikoNet (described in the supplemental material) has been modified to parameterize $f_\theta(x)$. This EikoNet is pretrained with samples from the prior $p(x)$ as input and the simulated travel time measurements as output. We use Adam as the optimizer [36] with a batch size of 32 and an E-step learning rate of 1e-3 and M-step learning rate of 5e-5. Hyperparameters $(\lambda_T, \lambda_V, \lambda_\theta)$ were empirically chosen by inspecting the loss on a grid search over hyperparameters. Results presented have been run with 10 EM iterations, each with 800 "E"-step epochs and 2000 M-step epochs. Each DeepGEM model takes ~6 hours on a NVIDIA Quatro RTX 5000.

### 4.3  Results

**Comparison to Baseline Approaches:**  We compare results from DeepGEM to results obtained using a baseline run by a seismologist. This iterative baseline alternates between source inversion and straight ray tomography, and is the standard approach used for blind tomography. Further detail on this baseline is provided in the supplemental material. The gradient model shown in Fig. 2 is used to perform the initial source inversion for this baseline; nonetheless, we find that the solution quickly diverges. Therefore, we rely on the expertise of the domain expert to decide when to terminate the optimization. As seen in Fig. 3, DeepGEM consistently outperforms this human-in-the-loop baseline across all source configurations. In Fig. 3 we also compare with a $\text{MAP}_{\theta,x}$ solution. $\text{MAP}_{\theta,x}$ is consistently outperformed by the DeepGEM $\text{MAP}_\theta$ approach across all source configurations. Additional results highlighting the sensitivity of $\text{MAP}_{\theta,x}$ to hyperparameters and initialization are shown in the supplemental material.

**Sensitivity to $\mathcal{L}_\theta$ Prior Choice:**  We evaluate DeepGEM's recovery of the true velocity structure shown in Fig. 1 using one of three different $\mathcal{L}_\theta$ priors shown in Fig. 2: homogeneous, gradient, and layer, as well as $\mathcal{L}_\theta = 0$. The homogeneous model takes on value of 6.419 km/s, the average velocity value of the true velocity structure. The gradient captures the increasing velocity as depth increases, and the layer model represents the true model without the added anomaly. As shown in Fig. 3, the mean squared error tends to decrease with the gradient and layer $\mathcal{L}_\theta$ prior models, which are closer to the true velocity structure.

**Sensitivity to Source Configuration:**  We evaluate results with sources that are both uniformly and randomly spaced throughout the region of interest. Improved performance is expected when the number of sources is increased and/or when sources are well distributed. To better understand DeepGEM's performance we introduce an ablation, where the velocity structure is learned by training $f_\theta(\cdot)$ with access to perfect source locations $x$ (i.e., $p(x)$ is a delta function). As is shown in Fig 3, even when training with true source locations, the anomaly is not well resolved until ~49 sources.

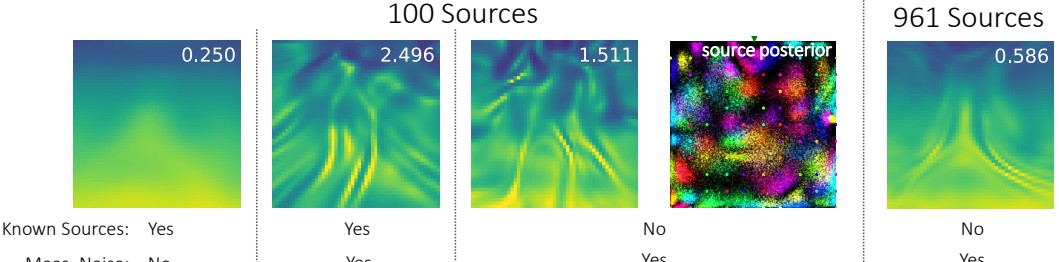

| Known Sources: | Yes | Yes | No | No |
|---|---|---|---|---|
| Meas. Noise: | No | Yes | Yes | Yes |

Figure 5: **DeepGEM recovery with a single receiver.** Velocity reconstructions shown in columns 3 and 5 demonstrate that DeepGEM is able to learn some of the true velocity features (see Fig. 1(a)), even when limited to measurements from a single receiver. However, these reconstructions show clear signs of overfitting to the measurement data. This is demonstrated by observing that the DeepGEM reconstruction with perfectly known source locations is significantly better with a single receiver when no measurement error is included on the measurements (comparing columns 1 and 2). Note that the learned posterior in column 4 appears to be non-Gaussian, but not multimodal.

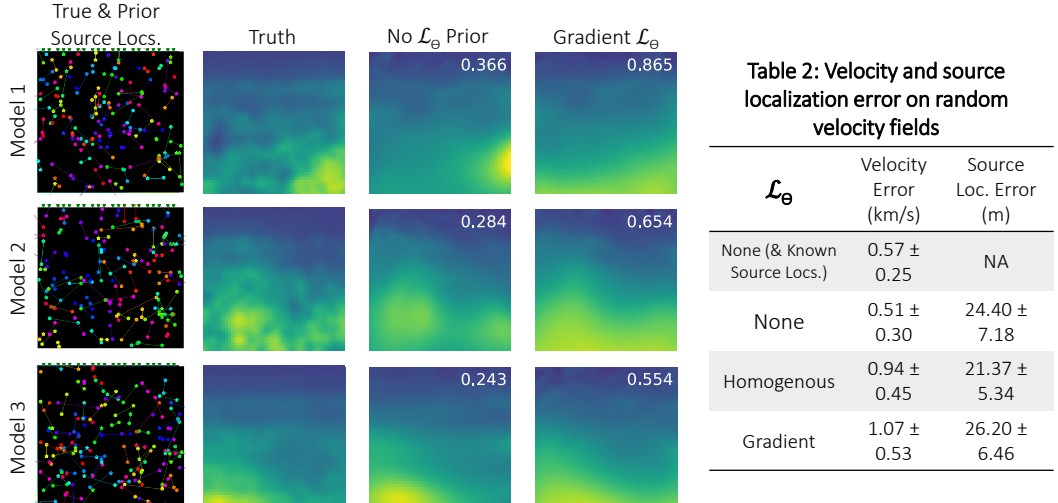

Table 2: Velocity and source localization error on random velocity fields

| $\mathcal{L}_\Theta$ | Velocity Error (km/s) | Source Loc. Error (m) |
|---|---|---|
| None (& Known Source Locs.) | 0.57 ± 0.25 | NA |
| None | 0.51 ± 0.30 | 24.40 ± 7.18 |
| Homogenous | 0.94 ± 0.45 | 21.37 ± 5.34 |
| Gradient | 1.07 ± 0.53 | 26.20 ± 6.46 |

Figure 6: **Performance of DeepGEM recovery on random velocity fields.** Ten random velocity fields were drawn from a GRF-based distribution and used to simulate travel time measurements with 20 receivers and 100 randomly placed sources. (left) Reconstructions obtained for 3 of these configurations are shown. (right) A table lists the mean and standard deviation of velocity and source error obtained across the ten models, each recovered using different $\mathcal{L}_\theta$ priors.

Fig. 3 shows one realization of DeepGEM results obtained from different counts of uniformly spaced sources. As expected, the MSE between the reconstructed and true velocity structure tends to decrease as the number of sources increases. Fig. 4 shows results obtained using five randomly generated source configurations, with 49 sources each. These results demonstrate that, although the reconstructed velocity structure is somewhat sensitive to the underlying source configuration, the primary features of the true velocity can still be recovered in all cases. Velocity and source localization error obtained for different random configurations are shown in the accompanying table.

**Sensitivity to Number of Receivers:** In the case of a single receiver, there exists an entire ring of source locations that result in the same travel time measurement. Fig. 5 contains results from this challenging one-receiver setting using DeepGEM with/without measurement noise and with/without known source locations. Since there is only one receiver, the velocity model is able to easily overfit. However, perhaps surprisingly, artifacts are more severe when source locations are perfectly constrained. These artifacts are caused by the velocity model overfitting to noise in the travel time measurements, and are substantially reduced when noise-free travel time measurements are used. Note that the recovered source location posterior $q_\phi(x)$ obtained by the "E"-step is non-Gaussian.

**Sensitivity to Velocity Structure:** In Fig. 6, DeepGEM is tested on randomly generated velocity fields, each generated from a Gaussian random field (GRF) described in the supplemental material.

These results show that DeepGEM works well at recovering the primary features across a variety of different velocity structures. As compared to when $\mathcal{L}_\theta = 0$, the $\mathcal{L}_\theta$ gradient prior biases the reconstruction towards smoother velocity structures. The accompanying table contains error statistics for ten different randomly generated velocity fields.

## 5    Case study: blind deconvolution

We apply DeepGEM to the problem of blind deconvolution to further demonstrate the generality of our approach. Blind deconvolution is a classic ill-posed imaging problem that aims to reconstruct a sharp image from a blurry image with an unknown PSF [19, 14, 25, 21, 22, 20, 6, 1, 27]. Blurry images, caused by handheld camera shake, can be modeled using a single spatially-invariant blur kernel:

$$y = x * k_\theta + \varepsilon \quad \text{for} \quad \varepsilon \sim \mathcal{N}(0, \sigma), \tag{8}$$

where $y$ is the blurry image, $*$ represents a 2D convolution, $x$ is the true sharp image, $k_\theta$ is the spatially invariant blur kernel, and $\varepsilon$ is additive Gaussian noise.

### 5.1    DeepGEM setup for blind deconvolution

For blind deconvolution, we parameterize the forward model $f_\theta(\cdot)$ using a deep network consisting of multiple convolution layers without non-linear activation, as proposed in [3]. Multiple convolutional layers without activation simply overparameterizes a linear blur kernel, which has been empirically shown to produce multiple good minima that are easier to converge to. To ensure the blur kernel is non-negative and volume preserving, we use a Softmax layer and normalize the kernel. For an $n$ layer network with weights $\theta_i$ for $i = 1, ..., n$, the resulting parameterized forward model is:

$$f_\theta(x) = x * \hat{k}_\theta = x * \left[ \frac{\text{Softmax}(\theta_1 * \theta_2 * ...\theta_n)}{\|\text{Softmax}(\theta_1 * \theta_2 * ...\theta_n)\|_1} \right] \tag{9}$$

| Measured Blurry Image | True Sharp Image and Measured Blur Kernel | Reconstructed Image using Measured Blur Kernel | Reconstructed Image and Blur Kernel using DeepGEM |

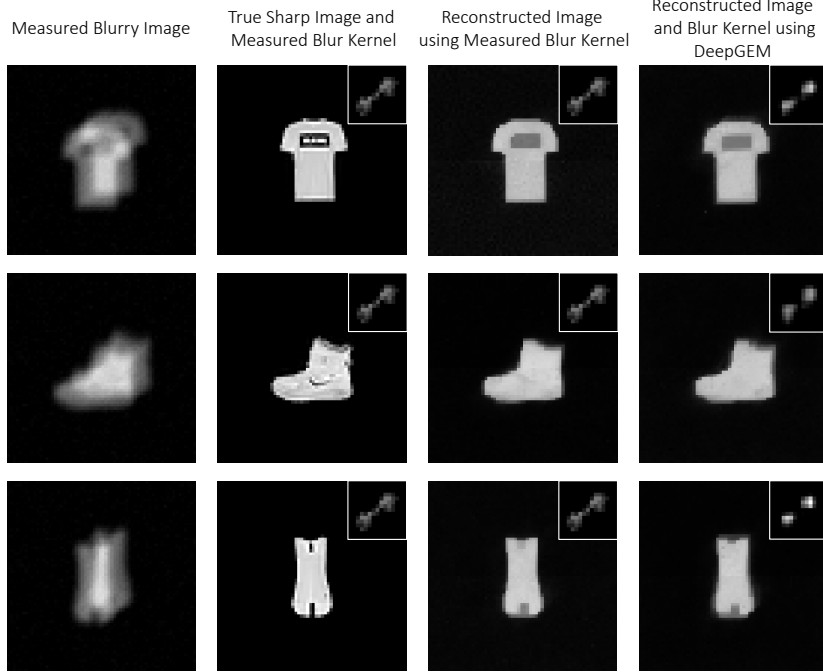

Figure 7: **DeepGEM results are comparable to inversion with known blur kernels for simple Fashion MNIST images.** Blurry measured images (column 1) are generated using the true sharp image and blur kernel (shown in column 2). The deconvolved images obtained using the true blur kernel are shown in column 3. Reconstructed deconvolved images and corresponding inferred blur kernel from DeepGEM are shown in column 4.

**Implementation details:** We demonstrate DeepGEM using simple Total Variation (TV) regularization in place of $\log p(x)$. We assume a Gaussian prior on the noise on the blurry measurements where $\varepsilon \sim \mathcal{N}(0, 0.01)$ as well as a sparsity prior on the reconstructed kernel through an $\ell_{0.8}$ soft constraint.

The posterior distribution of the sharp image, $x$, is estimated using a Real-NVP network $G_\phi(\cdot)$ with 16 affine coupling layers. We use 5 convolution layers to parameterize $k_\theta$. We use Adam as the optimizer [36] with a batch size of 64 and an E-step learning rate of 5e-4 and M-step learning rate of 1e-4. Hyperparameters, weights used for sparsity and TV priors, were empirically chosen by a grid search over hyperparameters. Results presented have been run with 10 EM iterations, each with 400 "E"-step epochs and 400 M-step epochs, which takes $\sim$1 hour on a NVIDIA Tesla V100.

## 5.2 Results

In Fig. 7 we show results from DeepGEM on three different Fashion-MNIST [34] images. The Fashion MNIST images are $64\times64$ pixels in size and each blur kernel is contained within $15\times15$ pixels. We choose to use images from Fashion-MNIST because they are compatible with the TV prior used in place of $\log p(x)$. In contrast, other natural images such as images from [22] (which were originally curated for deconvolution) are not compatible with this handcrafted prior. While this highlights that handcrafted priors such as TV are often insufficient, we choose to leave this to future work. Please refer to the supplemental material for more details about the performance of a TV prior on different types of images.

The blurry images in the first column of Fig. 7 exhibit artifacts from the blur kernel in the form of repeated features, caused by the bi-modal blur kernel. The images recovered via DeepGEM are much sharper and are comparable to the images reconstructed with the known blur kernel. Additionally, DeepGEM reconstructs blur kernels (i.e., $k_\theta$) that are similar to the true kernel shape, with two lobes along the same diagonal. Note that even when there is perfect knowledge about the shape of the blur kernel, detailed structures on the images are not recovered due to the TV prior.

## 6 Conclusions

In this paper we present DeepGEM, a deep probabilistic framework for tackling blind inverse problems through estimation of the forward model. DeepGEM achieves strong performance in the task of joint seismic tomography and earthquake source localization, substantially outperforming standard approaches currently being used in seismology on synthetic data. The proposed framework is flexible and can be applied to different applications that require estimation or fine tuning of forward model parameters. We demonstrate this flexibility by also applying the approach to a simple, but challenging, blind deconvolution problem. Future work includes applying this method to real seismic data, extending to other applications, and incorporating data-driven priors. Our results highlight the benefit of blending physically sound model-based techniques with learning machinery for blind inverse problems.

**Broader Impact**

DeepGEM can be used to solve for a system's model mismatch, which can then help improve our understanding of complex physical systems. However, this tool is not trustworthy enough for safety-critical systems. Nonetheless, this approach can benefit society through a better understanding of fundamental science and advanced earthquake prediction models via seismic imaging.

**Acknowledgments and Disclosure of Funding**

This research was carried out at the Jet Propulsion Laboratory and the California Institute of Technology under a contract with the National Aeronautics and Space Administration and funded through the President's and Director's Research and Development Fund (PDRDF). This work was sponsored by Beyond Limits, Jet Propulsion Laboratory Award 1669417, NSF Award 2048237, and generous gifts from Luke Wang and Yi Li. Additionally, we would like to thank He Sun for many helpful discussions. We declare no competing interests.

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
