# DeepGEM: Generalized Expectation-Maximization for Blind Inversion

**Angela F. Gao**[1]    **Jorge C. Castellanos**[2]

**Yisong Yue**[1]    **Zachary E. Ross**[2]    **Katherine L. Bouman**[1]

[1]Computing and Mathematical Sciences, California Institute of Technology
[2]Caltech Seismological Laboratory, California Institute of Technology
{afgao, jcastellanos, yyue, zross, klbouman} @ caltech.edu

## Contents

## 1  Related works for blind inversion

Similar works that use expectation maximization (EM) based deep learning approaches are usually specific to a single task, often times image classification. These similar works typically assume that

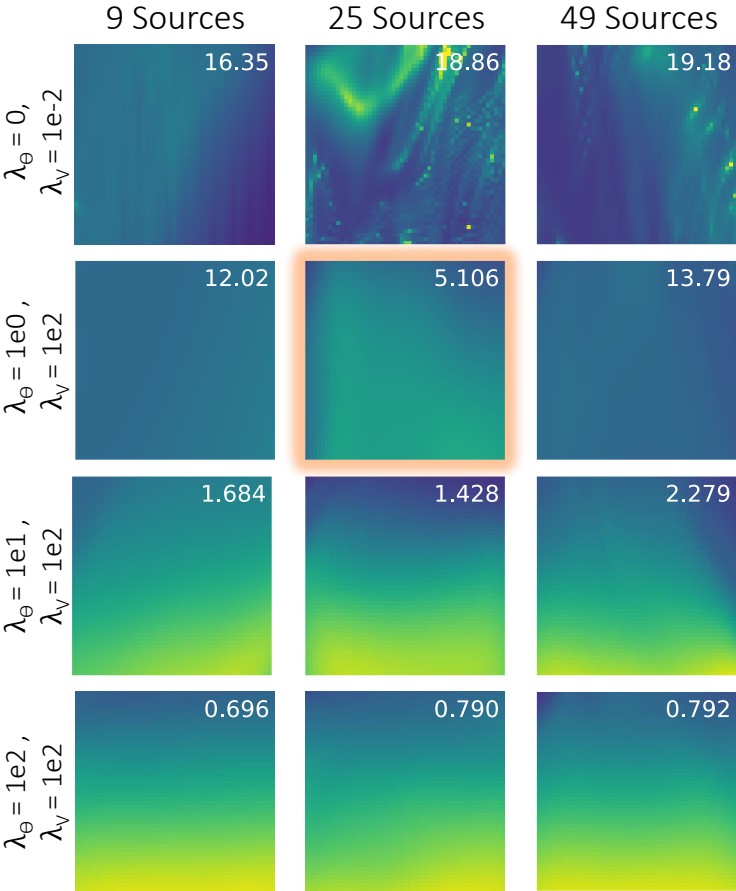

Figure 1: **Velocity reconstructions from MAP$_{\theta,x}$ with varying $\lambda_\theta$ and $\lambda_V$ prior weights.** Each column corresponds to the reconstructions obtained from a single noisy observation of travel times using MAP$_{\theta,x}$, where the true Earth velocity is shown in Fig. 1(a) from the main paper. Each row's reconstruction corresponds to different $\lambda_\theta$ and $\lambda_V$ values corresponding to varying strengths of the gradient $\mathcal{L}_\theta$ and $\mathcal{L}_V$ priors. Results shown are simulated using 20 surface receivers and a varying number of sources (9, 25, and 49) in a uniform grid. Note that as the prior $\lambda_\theta \mathcal{L}_\theta$ increases, the closer the velocity reconstruction becomes to the gradient prior model shown in Fig. 2 from the main paper. The velocity reconstruction MSE is included in the top right of each reconstruction. The model with the highest data likelihood is highlighted in orange.

the latent representation is Gaussian. Deep GMM [10] is an EM based approach that assumes the latent space is well modelled with a mixture of Gaussian distributions. Although this assumption works well for image classification, it does not generalize to other applications where the latent space is non-Gaussian (seen in the case of seismic tomography with only a few sensors). DeepEM [13] is another EM algorithm that uses deep learning machinery. DeepEM is a semi-supervised method and uses labelled data to approximate the posterior. However, rather than solving for the exact posterior, it approximates the posterior with a Half-Gaussian distribution determined through sampling Faster R-CNN.

Using alternative loss functions instead of the standard $\ell_2$ is another way to accommodate model mismatch into inversion. For example, Wasserstein distance is a loss function that has been used in many applications related to optimal transport, from seismology [12] to image reconstruction [11]. Since the dual of the Wasserstein distance [1] is differentiable, we may also be able to use it to better approximate $p(y|x)$ in the future.

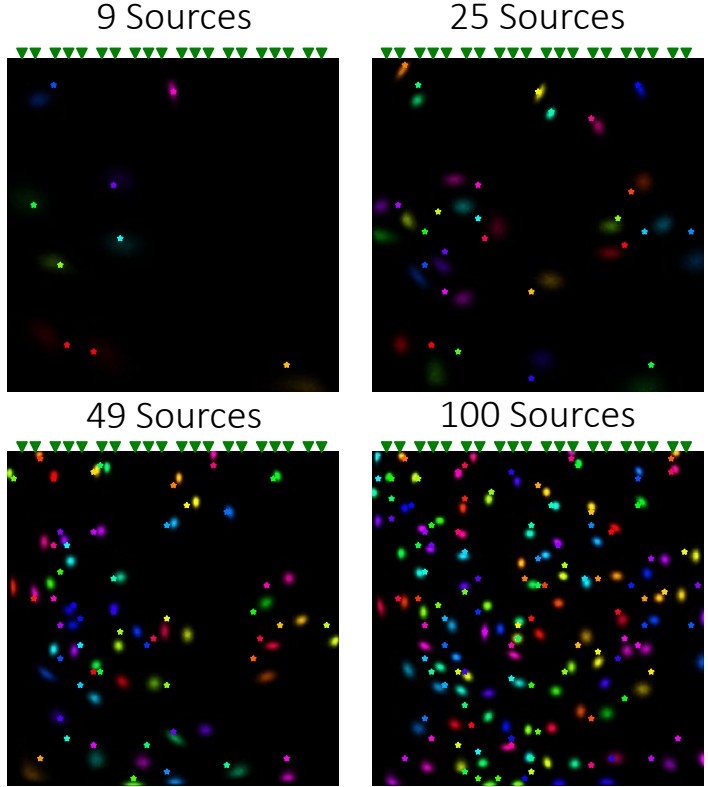

Figure 2: **Posterior visualization with varying number of random sources.** Results shown are simulated using 20 surface receivers and a varying number of sources (9, 25, 49, and 100) that are randomly sampled, with the legend shown in Fig. 2 of the main paper. Note that as the depth increases, the inferred posterior distribution has more uncertainty.

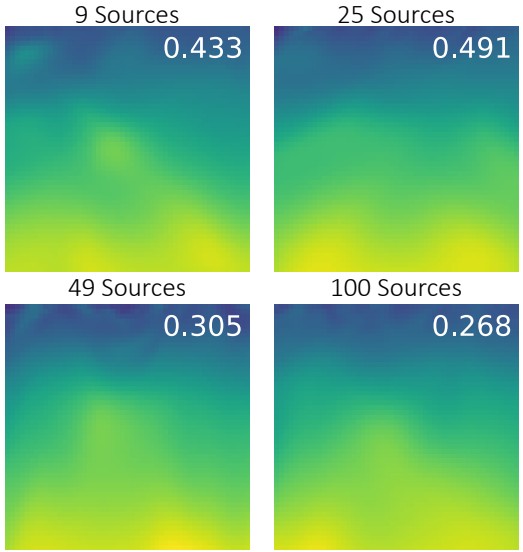

Figure 3: **Velocity reconstructions corresponding to the sources reconstructed in Fig. 2.** Results shown are simulated using 20 surface receivers and a varying number of sources (9, 25, 49, and 100) that are randomly sampled. The velocity reconstruction MSE is included in the top right of each reconstruction, where the true Earth velocity is shown in Fig. 1(a) of the main paper. Note that as the number of sources increase, the MSE tends to improve. Additionally, even with 9 sources, the anomaly is able to be reconstructed.

## 2 Blind seismic tomography

### 2.1 EM derivations

#### 2.1.1 "E"-Step derivation

We include expanded derivations of Section 3.1 Eq. 5 from the main paper. Here $q_\phi(x)$ represents a flexible variational distribution that well approximates the posterior distribution $p(x|y, \theta^{(t-1)})$. A normalizing flow network, $G_\phi(z)$, has input $z \in \mathbb{R}^{|x|}$, where $x = G_\phi(z) \sim q_\phi(x)$ when $z \sim \mathcal{N}(0, \mathbb{1})$ and $|x|$ is the dimension of $x$. Normalizing flow networks allow for exact computation of the log-likelihood $\log q_\phi(x)$, which is needed to solve Eq. 5. Eq. 5 can be derived using the variational objective to get

$$
\begin{aligned}
\phi^{(t)} &= \arg\min_\phi \mathrm{KL}(q_\phi(x)||p(x|y, \theta^{(t-1)})) \\
&= \arg\min_\phi \mathbb{E}_{x \sim q_\phi(x)} \left[ \log q_\phi(x) - \log p(x|y, \theta^{(t-1)}) \right] \\
&= \arg\min_\phi \mathbb{E}_{x \sim q_\phi(x)} \left[ -\log p(y|\theta^{(t-1)}, x) - \log p(x) + \log q_\phi(x) \right] \\
&\approx \arg\min_\phi \frac{1}{N} \sum_{n=1}^N [-\log p(y|\theta^{(t-1)}, x_n) - \log p(x_n) + \log q_\phi(x_n)] \\
&\text{for} \quad x_n = G_\phi(z_n), \quad z_n \sim \mathcal{N}(0, \mathbf{1}),
\end{aligned}
\tag{1}
$$

for a batch size of $N$ where $\log p(x)$ is a prior on the source and $\log p(y|x, \theta^{(t)})$ is the data likelihood. When assuming the measurements $y$ experience *i.i.d* additive Gaussian noise with standard deviation $\sigma_y$, $\log p(y|\theta^{(t)}, x_n) = \frac{1}{2\sigma_y^2} \|y - f_{\theta^{(t)}}(x_n))\|^2 + c$.

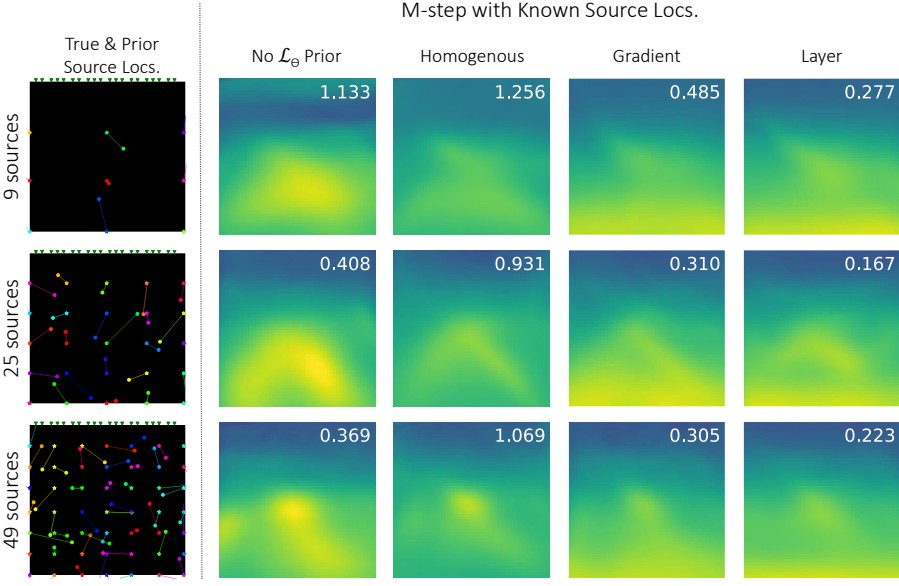

Figure 4: **M-step solutions (with known source locations) improve with more sources and better $\mathcal{L}_\theta$ prior models.** Each row corresponds to the reconstructions obtained from a single *noisy* observation of travel times, where the true Earth velocity is shown in Fig. 1(a) in the main paper. Results shown are simulated using 20 surface receivers and a varying number of sources (9, 25, and 49) in a uniform grid. Columns 2-5 show M-step results with known source locations obtained using different $\mathcal{L}_\theta$ priors. Note that results tend improve as the $\mathcal{L}_\theta$ prior becomes closer to the true velocity structure and as the number of sources increases. Note that these reconstructions tends to be blurrier than those shown in Fig. 5 due to the presence of Gaussian noise in the receiver measurements.

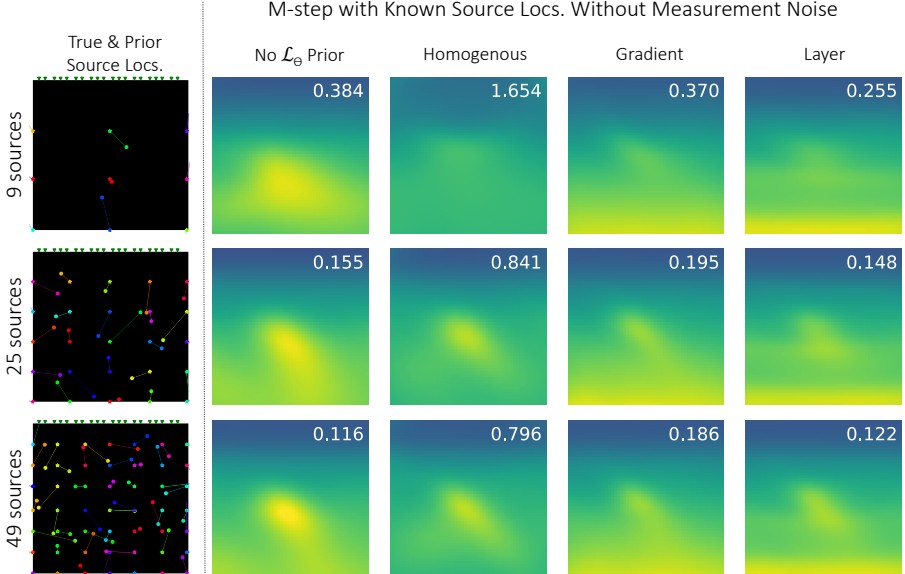

Figure 5: **M-step solutions (with known source locations) from noise-free measurements improve with more sources and better $\mathcal{L}_\theta$ prior models.** Each row corresponds to the reconstructions obtained from a single observation of *noise-free* travel times, where the true Earth velocity is shown in Fig. 1(a) in the main paper. Results shown are simulated using 20 surface receivers and a varying number of sources (9, 25, and 49) in a uniform grid. Columns 2-5 show M-step results with known source locations obtained using different $\mathcal{L}_\theta$ priors. Note that results tend improve as the $\mathcal{L}_\theta$ prior becomes closer to the true velocity structure and as the number of sources increases.

### 2.1.2  M-Step derivation

We include expanded derivations of Section 3.2 Eq. 6 from the main paper. The parameterized approximate posterior distribution, $q_{\phi^{(t)}}(x)$, is used from the "E"-step to update $\theta$, the parameters of the unknown forward model $f_\theta(\cdot)$. This is achieved by sampling from the learned normalizing flow network, $G_{\phi^{(t)}}(\cdot)$, to stochastically solve:

$$\theta^{(t)} = \arg\max_\theta \left[\log p(\theta|y)\right]$$

$$= \arg\max_\theta \left[\log p(y|\theta) + \log p(\theta)\right] \tag{2}$$

$$= \arg\max_\theta \left[\log\left(\int_x p(y|\theta,x)p(x)dx\right) + \log p(\theta)\right] \tag{3}$$

$$= \arg\max_\theta \left[\mathbb{E}_{x\sim p(x|y,\theta^{(t-1)})}\left[\log p(y|\theta,x)\right] + \log p(\theta)\right] \tag{4}$$

$$\approx \arg\max_\theta \left[\frac{1}{N}\sum_{n=1}^N \left[\log p(y|\theta,x_n)\right] + \log p(\theta)\right] \quad \text{for} \quad x_n = G_{\phi^{(t)}}(z_n), \; z_n \sim \mathcal{N}(0,\mathbb{1}), \tag{5}$$

where $p(\theta)$ is a prior on the forward model. This prior can be used to encourage the forward model parameters to remain close to an initial model $\tilde\theta$ by defining $\log p(\theta) \propto ||\theta - \tilde\theta||_2 + c$.

### 2.2  Baselines

### 2.2.1  Iterative straight ray baseline

The tomography baseline that we use is a straight ray tomography method based on [2]. The source localization baseline is the *maximum a posteriori* (MAP) solution given by

$$\hat{x} = \arg\min_x \sum_r \frac{1}{2\sigma_y^2}||T(x^*,r) - \hat{T}(x,r)||^2 - \log p(x) \tag{6}$$

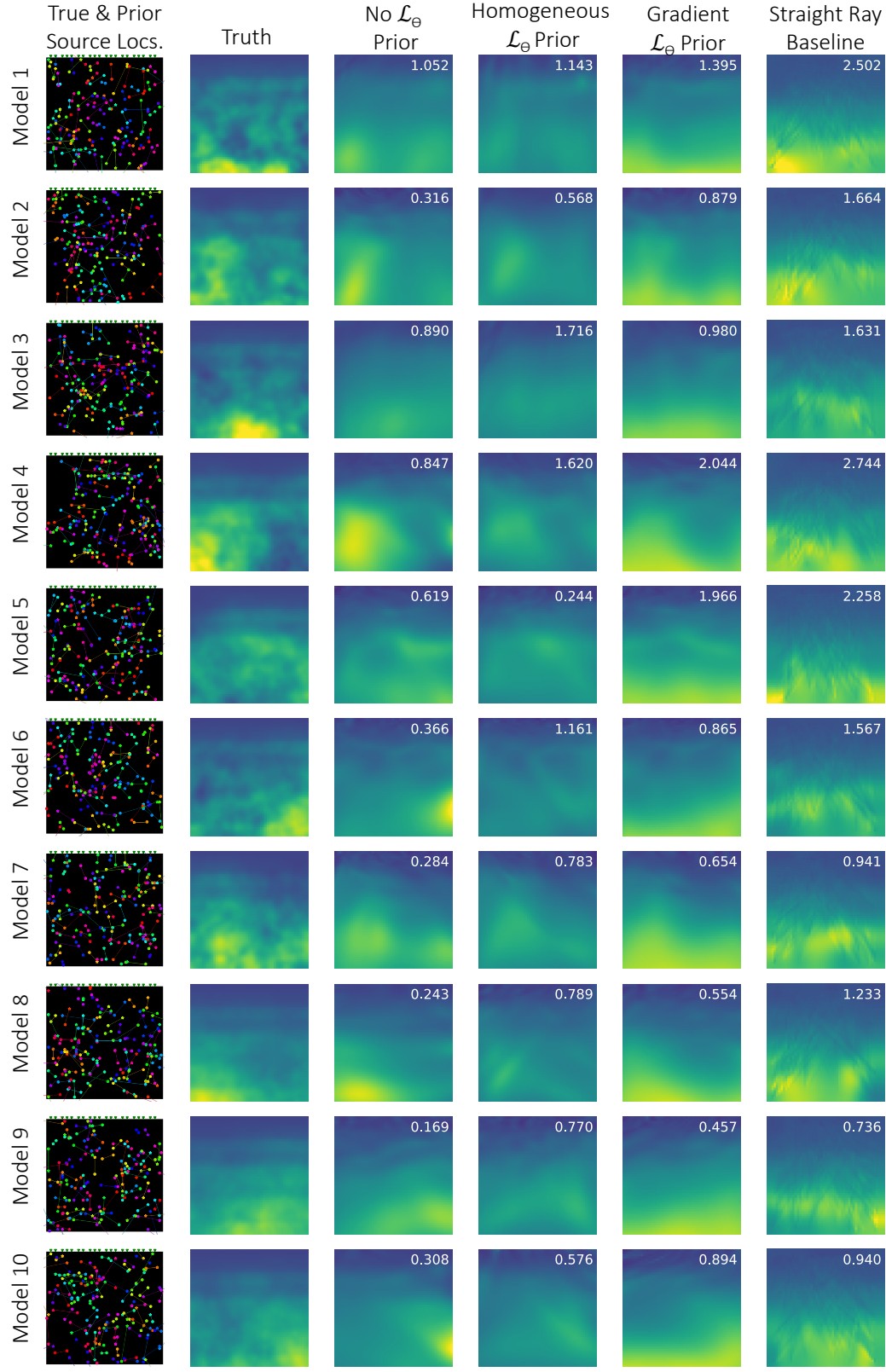

Figure 6: **Results from ten different GRF velocity models.** Ten random velocity fields were drawn from a GRF-based distribution and used to simulate travel time measurements with 20 receivers and 100 randomly placed sources. Reconstructions obtained for all of these configurations are shown. Velocity structure recovered using different $\mathcal{L}_\theta$ priors. The straight ray baseline is included.

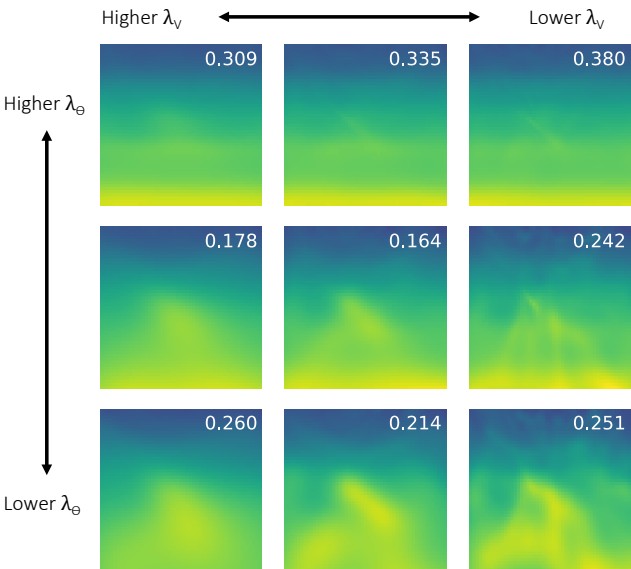

Figure 7: **Velocity reconstruction with varying prior parameters** $\lambda_\theta$ **and** $\lambda_V$. Each reconstruction is obtained from a single noisy observation of travel times, where the true Earth velocity is shown in Fig. 1(a) from the main paper. Each row corresponds to different $\lambda_\theta$ values corresponding to varying strengths of the layers $\mathcal{L}_\theta$ prior. Each column corresponds to different $\lambda_V$ values corresponding to varying strengths of the velocity invariance prior. $\lambda_\theta$ with values 1e-5, 1e-6, 1e-7 and $\lambda_V$ with values 1e-4, 1e-5, 1e-6 are used. The optimal parameters ($\lambda_\theta = $ 1e-6 and $\lambda_V = $ 1e-5) are used for all results shown in the paper. Results shown are simulated using 20 surface receivers and 100 sources in a uniform grid. The velocity reconstruction MSE is included in the top right of each reconstruction.

where $\sigma_y$ is the standard deviation of the measurement error, $T(x^*, r)$ is the true travel time measurement generated from the true source $x^*$ and receiver $r$, $\hat{T}(x, r)$ is the travel time measurement from an assumed velocity model and test source $x$, and $\log p(x)$ is the prior on the source location. We assume a Gaussian prior on the source where $x^* \sim \mathcal{N}(\bar{x}, \sigma_x)$. We discretize the possible source positions $x$ for efficiency.

The baseline solution results from alternating between the source localization and tomography to perform joint source and velocity inversion. The source baseline is initialized using a 1D gradient velocity model shown in Fig. 1(a) from the main paper. The iterative method usually converges when there is low uncertainty of the source location even with high uncertainty in the velocity model. However, when the 1D gradient velocity model poorly approximates the true velocity model, the iterative solution diverges. This is the case for the 1D gradient initialization. Thus, we chose to show only one iteration for our baseline results, a choice made by an expert seismologist with experience using the technique.

### 2.2.2 MAP$_{\theta,x}$

We show MAP$_{\theta,x}$'s sensitivity to $\mathcal{L}_\theta$ and $\mathcal{L}_V$ in Fig. 1. Note that with a stronger prior where $\lambda_\theta = $ 1e2 and $\mathcal{L}_V = $ 1e2, the reconstruction converges to the gradient prior model. With a weaker prior where $\lambda_\theta = 0$ and $\mathcal{L}_V = $ 1e-2, the reconstruction is far from the true underlying velocity, and is very sensitive to initialization choices. The model with the highest data likelihood is highlighted in orange. Note that this model does not have the lowest velocity reconstruction MSE.

### 2.3 Gaussian random field

The Gaussian Random Field (GRF) based velocity models were sampled such that:

$$V = 1.3(G + 0.5) * R + 2 \tag{7}$$

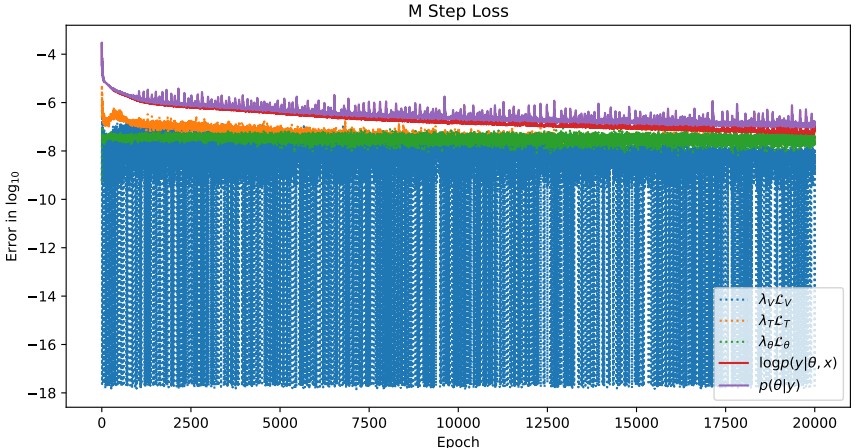

Figure 8: **Example loss curve from M-step only with known sources.** Noisy measurements are generated from 961 sources and 20 receivers. Total loss is plotted in purple.

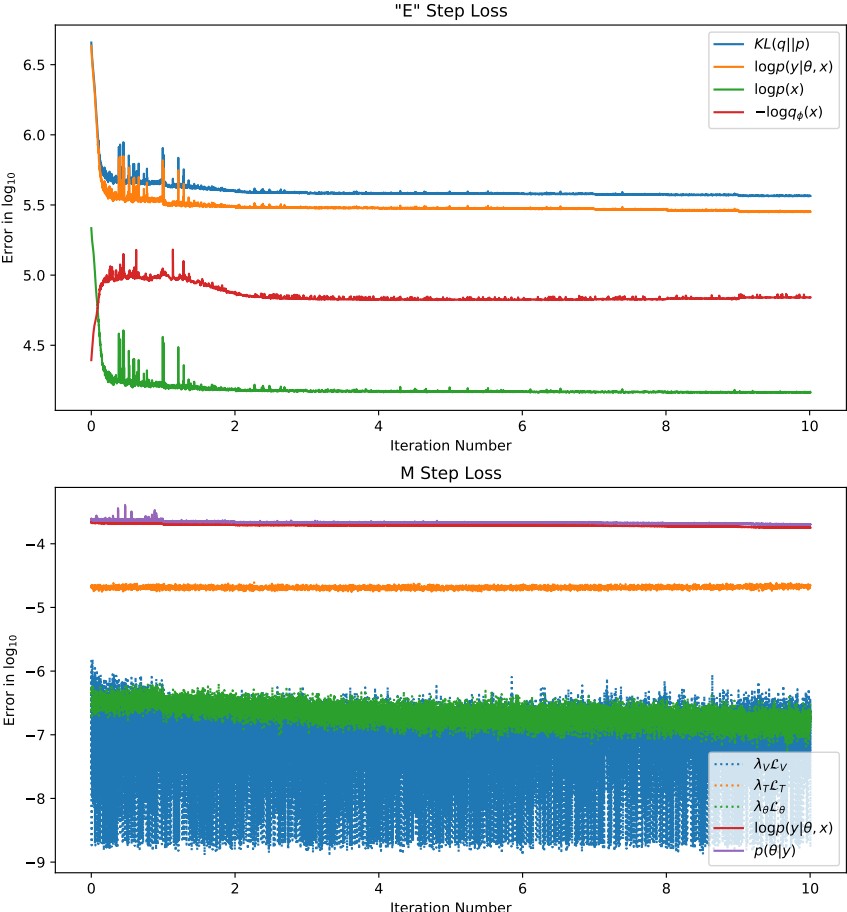

Figure 9: **Example loss curves for both "E" and M steps for DeepGEM reconstruction with 961 sources and 20 receivers.** Top plot shows the loss curves from the "E"-steps with the full loss plotted in blue. Bottom plot shows the loss curves from the M-steps with the full loss plotted in purple. Note that initially the $-\log q_\phi(x)$ term is increasing due to the data likelihood term taking over in optimization at early iterations.

where $G$ is a sample from a zero-mean Gaussian random field (with a von Karman covariance function that has parameters for correlation length, standard deviation, and the Hurst exponent), $*$ is element-wise multiplication, and $R$ is a 1D velocity model with layers.

### 2.4 Forward model parameterization

The EikoNet [9] is used to parameterize $f_\theta(x)$ in the blind seismic tomography problem. The EikoNet architecture used has been modified to have four residual blocks with sine activation [8]. Instead of training EikoNet using velocity, as in [9], our results are trained using simulated travel time measurements.

### 2.5 Additional results

#### 2.5.1 Posterior visualizations

We show the visualized posterior along with true source locations for 9, 25, 49, and 100 random source locations and 20 fixed receivers in Fig. 2. As depth increases, the estimated posterior increases in size as is expected. The corresponding velocity reconstructions are shown in Fig. 3. The overall mean squared error (MSE) of the velocity reconstruction tends to increase as the number of sources increases, but this is source-receiver configuration dependent.

#### 2.5.2 M-Step only reconstructions with known sources

In Fig. 4 and Fig. 5, we show more M-step only reconstruction (with known source locations) as comparisons to Fig. 3 in the main paper, where the true Earth velocity is shown in Fig. 1(a) from the main paper. Fig. 4 shows velocity reconstructions from a set of noisy measurements for 9, 25, and 49 sources with different priors on the velocity structure. In most cases, the M-step results in Fig. 4 have a lower MSE than the EM results in Fig. 3 from the main paper. Fig. 5 shows M-step velocity reconstructions from noise-free measurements for 9, 25, and 49 sources with different priors on the velocity structure. The reconstructions in Fig. 5 are better than Fig. 3 in the main paper when comparing MSE, which is due to the absence of measurement noise.

#### 2.5.3 Full GRF results

Results from ten different GRF velocity models are shown in Fig. 6 each with a different random source configuration. DeepGEM reconstructions wihtout $\mathcal{L}_\theta$ prior and gradient $\mathcal{L}_\theta$ prior both outperform the straight ray baseline for all velocity models. The GRF anomaly is reconstructed in all reconstructions without $\mathcal{L}_\theta$ prior except for model 3, which does not have enough sources within the anomaly. DeepGEM reconstructions are able identify the anomaly much more consistently than the straight ray baseline.

#### 2.5.4 Regularization

In Fig. 7, velocity reconstructions with 100 sources and varying the hyperparameters are shown, where the true Earth velocity is shown in Fig. 1(a) from the main paper. $\lambda_\theta$ with values 1e-5, 1e-6, 1e-7 and $\lambda_V$ with values 1e-4, 1e-5, 1e-6 are used. The optimal parameters ($\lambda_\theta = $ 1e-5 and $\lambda_V = $ 1e-4) are used for all results shown in the paper.

#### 2.5.5 Loss curves

We show that our method converges for the M-step optimization with known sources, shown in Fig. 8, and the joint DeepGEM optimization for blind inversion, shown in Fig. 9. Note that these curves are plotted in $\log_{10}$ scale.

## 3 Blind deconvolution

### 3.1 Total Variation Prior

Traditional approaches to blind deconvolution often solve for a sharp target image while simultaneously solving for the kernel $k$, which represents the camera's point spread function (PSF) [3, 5, 6]. These techniques make use of regularizers, such as total variation (TV) regularization on the sharp image and $\ell_1$ sparsity on the kernel, to constrain the ill-posed inverse problem [4].

TV assumes that the image gradients follow a Laplacian distribution. While TV is commonly used for many different image processing problems [4, 6, 7], it is (perhaps surprisingly) not a good model

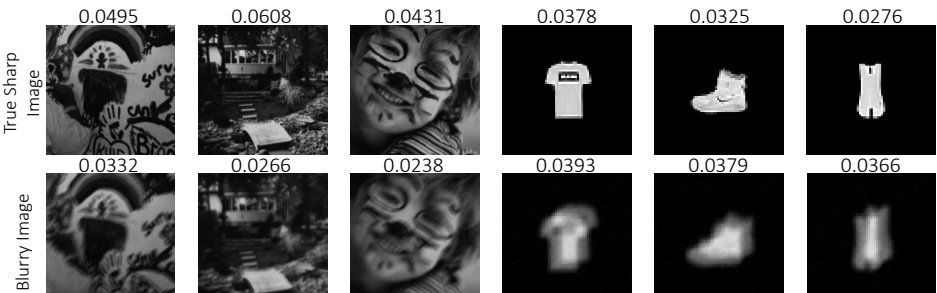

Figure 10: **Total variation (TV) values for the different true sharp images and blurry images used.** TV values for the true images (row 1) and blurry images (row 2). Note that the TV value is being used as $-\log p(x)$. Therefore a higher TV value indicates that an image is less likely.

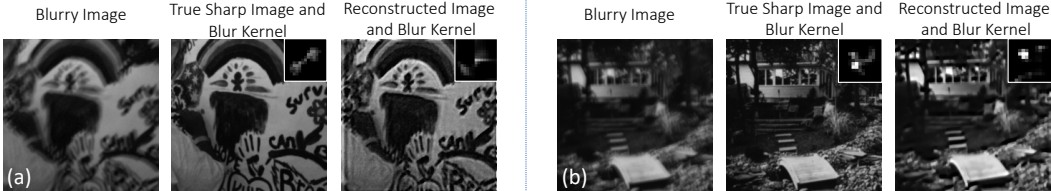

Figure 11: Blurry measured images (columns 1 and 4) generated using the true sharp image and blur kernel (shown in columns 2 and 5). Reconstructed sharp image and the corresponding inferred blur kernel from DeepGEM (columns 3 and 6).

to evaluate natural images as compared to blurry images. As described in [5], sparse image gradients prefer blurry images over than sharp natural images (first 3 columns). For the Fashion MNIST images TV priors prefer the sharp image over the blurry images. We show the TV values for both true and blurry images used in our experiments in Fig. 10; higher TV values indicates that the image is *less likely* since we define TV as $-\log p(x)$:

$$\text{TV}(I) = ||\nabla_x(I)||_1 + ||\nabla_y(I)||_1 \tag{8}$$

There are two ways to avoid preference towards the blurry image delta PSF solution: (1) better image priors and (2) over-estimating the uncertainty in the posterior conditional distribution during EM optimization. Both solutions can be easily incorporated in DeepGEM. Data-driven priors can easily be incorporated since our method only requires evaluation of the prior. We choose to leave this to later work.

## 3.2 Additional results

Here our goal is to highlight that a TV prior leads to poor deconvolution results for many natural images, unlike the results shown in Fig. 7 in the main paper. In Fig. 11, we show additional results for images from [5], which are of size $255 \times 255$. As described in the previous section, for these images the TV prior prefers the blurry image with an impulse kernel over the true sharp images.

The blurry images in Fig. 11 exhibit artifacts due to the shape of the blur kernel. In Fig. 11(a), the text in the deconvolved image is more legible, but there are still some ringing artifacts due to model mismatch. The reconstructed kernels roughly match the true kernels' shape with two lobes along the same diagonal. In Fig. 11(b), the edges in the blurry image are sharper in the reconstructed image. Although the reconstructed kernel is not located spatially at the same location as the true kernel, this does not significantly harm reconstruction; since the kernels are shift-invariant, the reconstructed image and learned kernel can both be shifted such that they reconstruct the same blurry image.

Note we suspect that the reason we converge to a sharper image than the initial blurry image is due to the fact that we initialize the blur kernel to a Gaussian kernel. The solution converges to a local minimum rather than the blurry image with impulse kernel explanation.