# OpenReview forum: "DeepGEM: Generalized Expectation-Maximization for Blind Inversion"
_NeurIPS.cc/2021/Conference — NeurIPS 2021 Poster_

### Official Review · Reviewer_FVfb · 2021-07-12

**Rating:** 8
**Confidence:** 4

**Summary:**

This paper proposes a general variational Expectation-Maximization framework aided by a normalizing flow generative network to solve inverse problems for which a forward physical model is known, but important model mismatch exists. The approach is evaluated on a blind seismic tomographic task and showed to quantitatively and qualitatively outperform a state-of-the art baseline through various experiments, that include ablation studies. The versatility of the framework is also illustrated by a small case-study on image blind deconvolution, where the authors report convincing qualitative results on two images.

**Limitations And Societal Impact:**

The authors correctly address the limitations and societal impact of their work.

**Main Review:**

This is a very good paper. The addressed topic is timely and relevant, the methodology is novel and technically sound, the paper is clearly written, experiments are well designed and the reported results are good and convincing. Some important clarifications are nevertheless required.

Some clarifications needed in section 3:
- In both eq. (5) and (6) the authors appear to be using an approximation of the expectation via a form of re-parameterization trick (as coined in the original VAE paper by Kingma and Welling), but this is not explicitly stated.
- Some derivations are missing to explain how the log-likelihood is "exactly computed" (line 131)
- More generally, it is not clear how (5) and (6) are solved in practice, some derivations should be provided.
- On lines 114-115, the authors explain that the MAP objective will be optimized using "only measurements y". It is not clear at this stage and only becomes clear later that a *single* measurement y is used. This is what allows the authors to remove the dependency on y from q_{\phi}[x) on line 127 and later (otherwise it does not make sense). Please clarify this.

Some clarifications needed in section 4:
- Lines 201-209: How many parameters do the RealNVP and EikoNet contain? How does this tie in with the number of available observations, which seem relatively small? More generally, did the authors use training/development sets (in particular to tune hyperparameters) and how were they design?
- For both Fig. 3 and 4: How do you explain that errors get lower as the number of sources increases? shouldn't it be the opposite, as the problem is less and less well-posed? Or is it because of the strong provided priors? please comment on that.
- Line 247: the authors write that the obtained posterior is non-Gaussian, but is it multimodal? (unclear from figure).

Typos:
-Line 87: can be used _to_ approximate
-Line 265: emperically -> empirically
-Line 294: by also applying the approach to the simple, but challenging, blind deconvolution problem -> to _a_ simple ...


**Time Spent Reviewing:**

2,5

---

> ### Author Response · Authors · 2021-08-10
> **Answer to FVfb**
>
> Thank you for your positive feedback and suggestions. Responses to each point are as follows:
>
> *"In both eq. (5) and (6) the authors appear to be using an approximation of the expectation via a form of re-parameterization trick...”*
>
> We are not using the reparameterization trick as we can use normalizing flows to compute exact log likelihoods of samples. This allows us to represent much more complicated posterior distributions than can be represented using the reparameterization trick.
>
> *"Some derivations are missing to explain how the log-likelihood is "exactly computed" (line 131)”*
>
> This has already been derived in normalizing flows papers that we have cited (Kingma & Dhariwa, Dinh et al.). We will clarify this in the text and include it in the supplemental material.
>
> *“More generally, it is not clear how (5) and (6) are solved in practice, some derivations should be provided.”*
>
> Solving equations 5 and 6 in practice depends on what distributions are assumed for both measurement error and priors. We have included how they are solved assuming i.i.d additive white Gaussian noise (line 135) as well as a forward model prior (line 142).
>
> *"On lines 114-115, the authors explain that the MAP objective will be optimized using "only measurements y". It is not clear at this stage and only becomes clear later that a single measurement y is used...”*
>
> Thank you for highlighting this confusion; we will clarify this in the revised manuscript. Unlike in standard VAE training, where each measurement is a different training point, in our formulation $y$ is static and contains measurements from all events together.
>
> *“Lines 201-209: How many parameters do the RealNVP and EikoNet contain? How does this tie in with the number of available observations, which seem relatively small? More generally, did the authors use training/development sets (in particular to tune hyperparameters) and how were they design?”*
>
> The number of parameters of the RealNVP model depends on the number of observations. For seismic tomography with 100 sources and 20 receivers there are 278K parameters. The number of parameters of the EikoNet is independent of the number of observations and has 3M parameters.
>
> *“For both Fig. 3 and 4: How do you explain that errors get lower as the number of sources increases? shouldn't it be the opposite...”*
>
> In tomography, having more sources results in more rays sampling the domain. Having better ray coverage provides more information about the wave speed at each location, making the tomography problem less ill-posed (usually resulting in a lower velocity error).  With fewer sources the optimization objective is also more non-convex, often leading to convergence to a worse local minima.
>
> *“Line 247: the authors write that the obtained posterior is non-Gaussian, but is it multimodal? (unclear from figure).”*
>
> In this example, it appears the posterior is not multimodal.

---

### Official Review · Reviewer_AwcU · 2021-07-14

**Rating:** 4
**Confidence:** 4

**Summary:**

This paper considers the problem of inverting a signal from a given forward model $y=f_\theta(x^*)+\epsilon$, while accounting model mismatch from true model $f_{\theta^*}$. The authors propose to solve this problem by first approximating the distribution of $x$ with $q_\phi(x) \sim G_\phi (z)$ using a deep model and carrying out an expectation over $x\sim q_\phi(x)$ of the log-likelihood $p(y|\theta,x)$. Model parameters $\theta$ are approximated via maximization over this expectation (EM framework). They call this framework DeepGEM.

The authors specifically apply this mathematical model to 1) blind seismic tomography, where the task is to estimate the velocity structure of earthquake waves. The velocity computation depends on travel time $T$ (this is the variable observed) parameterized by source location $x$ and receiver location $r$.  In this paper $T(x,s) = f_\theta(x,s) $ and $y \approx T(x,s=r)$. 2) Blind deconvolution with $y=x * k_\theta +\epsilon$.

**Limitations And Societal Impact:**

Limitations: Line 216: authors rely on domain expert to decide when to terminate the optimization for the baseline, without which the algorithm diverges. It's unclear how to gauge the accuracy of baseline.


**Main Review:**

The authors propose to use deep probabilistic models [31] to estimate a-posteriori distribution of signal x, in the context of inverse problems with model mismatch. The approach of using Expectation-Maximization has been used for solving inverse problems with model mismatch, such as blind deconvolution [20]; however this is the first paper to consider deep priors under this framework. Moreover the application of such approach to the domains of blind seismic tomography and blind deconvolution are novel. However there are several concerns.

It is unclear if this kind of approach will even converge; this paper will benefit from showing training curves showing reduction of loss value.

In Line 142, why is the prior on $\theta$ (remains close to initial) reasonable? What if the initialization is far from the true parameters $\theta^*$. This section requires a comment on how $\theta$ is initialized.

In Figures 3 and 4 what are the metrics reported in the top right corner of each figure?

For the blind deconvolution case study, there is no baseline comparison, even though baseline models have been referenced in literature survey (eg. [20,21]). Since this paper is purely experimental in nature, baseline comparisons are important.

Some notations need to be clarified or introduced as described below.

Some line specific comments are as below:

Line 102: define |x| to be dimension/cardinality

Step 3.1: explain what value theta assumes here. Is it randomly initialized?

Line 132: simplifying KL divergence -- N has not be defined anywhere so far.

Line 135: if y includes gaussian noise with 0 mean, then eps = || y - f(x) ||^2,

Line 142: why should theta stay close to \tilde(theta)? Why is this a reasonable assumption?

Line 185: $\bar{x}$ is not defined at this point (it is defined later at 195)

Line 187: What is $\tilde{V}(s)$ and how does it differ from $V(s)$?

Apart from these concerns the paper is otherwise well written in terms of introduction and explanations related to the blind seismic tomography.

**Time Spent Reviewing:**

6 hours

---

> ### Author Response · Authors · 2021-08-10
> **Answer to AwcU**
>
> Thank you for your feedback. Responses to each point are as follows:
>
> *“It is unclear if this kind of approach will even converge...”*
>
> Since this is an EM algorithm, our method borrows all of the inherent properties of EM, which includes convergence. However, as with EM, convergence to a global minimum is not guaranteed. We would be happy to provide training curves in the camera-ready manuscript or supplemental material .
>
> *“Line 142: why should theta stay close to $\tilde{\theta}$? Why is this a reasonable assumption?”* and *“Step 3.1: explain what value $\theta$ assumes here. Is it randomly initialized?”*
>
> Initialization on $\theta$: We initialize $\theta$ using maximum likelihood estimation assuming the source location is the mean of the source location prior.
>
> Prior on $\theta$: We have tested multiple priors on $\theta$ for different velocity structures (Figure 3 and 6 in main paper, Figure 5 in supplemental). In some cases, having a simple 1D velocity prior improves the resulting reconstruction, especially with few sources (9 sources). However, assuming closeness to a prior is detrimental to the reconstruction when the prior is far from the true model.
>
> *“In Figures 3 and 4 what are the metrics...”*
>
> These metrics are the mean squared error of the velocity reconstruction, as stated in the figure caption.
>
> *“For the blind deconvolution case study, there is no baseline comparison, even though baseline models have been referenced in literature survey (eg. [20,21])...”*
>
> This is a great point.  We actually found it difficult to conduct a fair comparison here, because baseline models use various heuristics to avoid sensitivity to a TV prior (see our supplemental material for results that demonstrate how a TV prior prefers blurry images).  Our goal with this set of experiments was simply to demonstrate the generality of our approach rather than compete with the state-of-the-art.  If reviewers feel that the paper would make a stronger contribution with the focus purely on blind seismic tomography, we would be willing to remove blind deconvolution from the main paper.
>
> *“Line 187: What is $\tilde{V}(s)$and how does it differ from $V(s)$?”*
>
> $\tilde{V}(s)$ is the assumed incorrect forward model, whereas $V(s)$ is the reconstructed velocity model.  The parameters of $\tilde{V}(s)$ are used for the mean of the prior $p(\theta)$.

---

### Official Review · Reviewer_M56A · 2021-07-16

**Rating:** 7
**Confidence:** 5

**Summary:**

This paper proposes a normalizing-flow-based framework to solve inverse problems with model mismatch. The main idea is to use a normalizing flow to model the posterior distribution of the unknowns x conditioned on the current model parameters \theta and the observation y, as part of the expectation-maximization algorithm. The proposed strategy yields results that seem to outperform "traditional" methods in seismic tomography and blind deconvolution.


**Limitations And Societal Impact:**

Yes.

**Main Review:**

Originality: The paper builds on existing ideas in an original way. It proposes a creative use of normalizing flows to solve inverse problems with model mismatch. On the other hand, the central idea---using flows to model conditional distributions---is not a new one (among other places, it is described in the cited DPI paper).

Quality: The paper is well written and the empirical results seem solid.

- One somewhat philosophical comment is that model mismatch and blind inverse problems (as in blind deconvolution) are not really the same thing. In blind inverse problems we are up front about the fact that some explicit parameter of the forward model is unknown and should be identified as part of the inverse problem. In vanilla model mismatch we don't always have a model for the mismatch, and we often assume that the mismatch is in some sense small. In your examples, \theta is really a bona fide unknown, like x.

- This is especially evident in that in your seismic tomography experiments (e.g. Figure 3) you make more effort to compare the wave speed models \theta than the source locations x which are ostensibly the object you're trying to solve for. There is nothing wrong with this but to me it indicates a certain arbitrariness in the split between the unknowns of interest and the model parameters. In this case for the split you leverage independence of x and \theta (Btw, is this verified in practice? I would expect that the sources are mostly located along some geological features which indicates a dependence between the geology \theta and the source locations. Or am I wrong here?). Of course, in seismic imaging the split is very natural and it corresponds to well-known inverse problems (tomography vs source localization), but one could also say that the model is perfectly known---it is the wave equation---and you're jointly solving for the parametric RHS and the wave speed.

- I am wondering about the "seismologist-run" baseline. Why limit oneself to straight rays? The main difference between the two approaches then seems to be in your use of the EikoNet rather than in the use of flows and the EM algorithm. But EikoNet could be easily used as the forward model in the alternating scheme used by the seismologist, without involving flows. Wouldn't that be a fairer comparison? Otherwise this is not just model mismatch (as in the wave speed being different) but also model class (or physics) mismatch, since regardless of the wave speed your method uses physical, bent rays, while the seismologist's method uses unphysical straight rays. I think that a solid comparison should also use EikoNet for the baseline.

- Somehow related to the above two comments: in several places in the manuscript you advertise the "generality" of the approach, but the generality seems to equate the fact that normalizing flows can model conditional distribution which was shown already in the DPI paper. Granted, your application is a bit different. To the contrary, for each inverse problem one wants to address (here tomography and deconvolution), one has to design a differentiable, parameterized forward model (here in both cases a neural net). Thus the proposed method is not an out-of-the-box general method. The bulk of the difficulty is, I expect, in the design of this differentiable forward model. I believe this is a point worth elaborating.

- Out of curiosity, how do you expect your method to scale? It seems that you need to train a normalizing flow at every iteration of the EM algorithm? How long does one iteration of that take? Can you speculate about performance on large-scale problems?

Clarity: The papers is very clear and well written.

Significance: The paper addresses an important problem of machine learning for inverse problems with model mismatch. It is surely a significant problem for the inverse problems / wave imaging community. The machine learning contribution on the other hand is minor (in view of the existing DPI and EikoNet papers), but inverse problems have been gaining traction at NeurIPS over the past couple of years.


**Time Spent Reviewing:**

3

---

> ### Author Response · Authors · 2021-08-10
> **Answer to M56A**
>
> We appreciate your feedback, and we address our main contribution, blind inversion vs. model mismatch, and baselines in our global comments. Responses to other points are below:
>
> *“... leverage independence of x and \theta (Btw, is this verified in practice? I would expect that the sources are mostly located along some geological features which indicates a dependence between the geology \theta and the source locations. Or am I wrong here?).”*
>
> This is a great point. In reality, source locations and velocity structures are often related. Many sources lie along faults, which have low wave speed and show up as zones of damaged rocks. However, there can be fault zones that do not have these signatures. In practice, these two problems are treated independently, which avoids such biases [A. Plesch, et. al, *Bulletin of the Seismological Society of America*, 2020].
>
>
> *“I am wondering about the "seismologist-run" baseline... I think that a solid comparison should also use EikoNet for the baseline.”*
>
> You’re absolutely right about that being a more direct comparison, and we actually do make this comparison. Please see the global comment on baselines.
>
>
> *“To the contrary, for each inverse problem one wants to address (here tomography and deconvolution), one has to design a differentiable, parameterized forward model ...”*
>
>
> It is true that the generality of this method is tied to having a differentiable forward model. In the case of seismic tomography and deconvolution we have a differentiable parameterized forward model, so we can solve for these parameters directly. However, when the forward model is unknown or non-differentiable one can simply solve for a network that acts as a surrogate to the forward model.  We also plan to explore ways to use non-differentiable parameterized forward models in future work.
>
>
> *“Out of curiosity, how do you expect your method to scale?”*
>
> In terms of compute time, the problem should scale linearly with respect to the number of measurements. In the current framework, a normalizing flow network is trained every EM iteration. Each iteration of the EM algorithm takes ~0.6 hours for 20 receivers and 100 sources (see line 208).

---

> > ### Comment · Reviewer_M56A · 2021-09-02
> > **improving score**
> >
> > After reading the author response I decided to improve my score by one point. The main reasons are the authors' willingness to rebrand their method as blind inversion and certain global clarifications about the novelty. That said there are still a few things I'm uncertain about:
> >
> > - The argument about independence of sources and geology to me seems to be precisely an argument for their _dependence_ (not every source will be along a fault but many will, more so than elsewhere, thus one can benefit by modeling it)
> > - Using a network as a surrogate for the forward model is not something that would obviously work: it would greatly increase the number of unknown degrees of freedom (there are many possible forward models). My gut feeling, quite easily wrong, is that the two addressed problems nail a sweet spot of blindness.

---

### Official Review · Reviewer_xa1o · 2021-07-21

**Rating:** 4
**Confidence:** 4

**Summary:**

This paper proposes to use variational EM algorithm to simultaneously recover forward model and latent variable in an inverse problem. The proposed method is extensively evaluated on seismic tomography.

**Limitations And Societal Impact:**

Have the authors adequately addressed the limitations and potential negative societal impact of their work? If not, please include constructive suggestions for improvement：Yes

**Main Review:**

- Novelty: the proposed method consists of using neural nets to parametrize forward models and use variational EM to solve for parameters. Both aspects are well established and does not provide much new insights.

- Experiments: the proposed method is extensively evaluated on blind seismic tomography. However this paper lacks quantitative results and comparison with baseline methods. In blind convolution there are no quantitative comparison. It seems to me that for publication purposes, this paper should be reorganized as a  technical novelty on seismic tomography instead of a general machine learning algorithm, as in its current form the evaluation is not comprehensive enough.

- Scope: this paper discusses model mismatch, however in blind deconvolution experiments, instead of using a mismatch model the authors treat the forward model as blind. To properly deal with model mismatch, the algorithm should be able to utilize the inaccurate initial model to improve performance. This aspect is not reflected in the deconvolution case study.

**Time Spent Reviewing:**

4 hours

---

> ### Author Response · Authors · 2021-08-10
> **Answer to xa1o**
>
> We appreciate your feedback, and we address our main contribution, blind inversion vs. model mismatch, and baselines in our global comments. Responses to other points are below:
>
> *“However this paper lacks quantitative results and comparison with baseline methods.“*
>
> We actually do show quantitative results for the blind seismic tomography problem in the form of MSE for both source localization error and velocity reconstruction. This is shown in Figures 3, 4, and 6 and Tables 1 and 2 of our paper, as well as Figures 1, 3, 4, and 5 of the supplementary material. Additionally, we show comparisons to two baselines, MAP and a common seismology-based straight ray baseline, in Figure 3 of our paper, as well as comparison to the straight-ray baseline in Figure 5 of the supplementary material.
>
> *“In blind convolution there are no quantitative comparison...”*
>
> These are good points and we agree that the deconvolution results are not as detailed as the seismic tomography results. Our goal with including blind deconvolution was to show the generality of the method. If the reviewers think it’s a stronger paper without blind deconvolution, then we can move it to the supplementary material.

---

### Author Response · Authors · 2021-08-10
**Global Comments**

We thank all the reviewers for their constructive feedback  and positive comments, noting the  paper as presenting a “novel”, “creative”, and “convincing” solution on a “significant problem for the inverse problems / wave imaging community”. We clarify here a few common points, and also respond to each reviewer individually.

**Primary Contributions**

Our work makes two significant contributions. First, it proposes a deep variational EM framework, which to the best of our knowledge has never been proposed before. Prior deep approaches typically involve VAEs, which solve for a variational solution, but do not perform deep variational EM and do not allow for complex posterior distributions on the fundamental latent variables (which represent the unknown sources). Unlike typical iterative coordinate descent optimization, our method borrows the guaranteed convergence properties of EM. Second, our work demonstrates a novel application to blind seismic tomography, significantly outperforming existing baselines and showcasing the value of deep variational EM. Unlike previous approaches in blind seismic tomography that are unstable to iterative optimization and can converge to bad local minima, our approach demonstrates stable optimization by making use of an inferred posterior distribution.  We will clarify these points in the revised manuscript.

**Blind inversion vs. model mismatch**

We agree with Reviewers xa1o and M56A that this paper is best described as a method for blind inversion rather than model mismatch. We will update the manuscript accordingly.

**Baseline clarification**

We chose our baselines after consultation with expert seismologists. The straight ray baseline is an established method in the field and is typically used for its simplicity and stability, allowing for linear inversion (see the last column of Figure 3 in the main paper and the last column of Figure 5 in the supplementary material). Additionally, we show a baseline with EikoNet as the forward model by solving for the $MAP_{x, \theta}$ solution in column 7 of Figure 3. However, $MAP_{x, \theta}$ is a challenging optimization problem and incredibly sensitive to priors and hyperparameters, as shown in Figure 1 of the supplementary material.

**Focus on Blind Seismic Tomography**

If requested, we are willing to move the blind deconvolution section to the supplemental material and focus the paper on contributions in seismic tomography.

---

> ### Comment · Reviewer_FVfb · 2021-08-24
> **Still a good paper**
>
> I read the other reviews and author responses.
>
> - I agree with the other reviewers that the paper might be best described as (semi)blind inversion rather than model mismatch, however I see this as a minor/philosophical issue
> - The evaluation of the paper is adequate to me: For seismology, expert knowledge is used for qualitative evaluation while simulations are used for quantitative evaluation. The blind image deconvolution part is clearly a minor addition to the paper that would require a separate study on its own for proper evaluation, but I still find insightful to include it, as it illustrates well the versatility of the framework
> - I have to disagree with the above general statement made by the authors: "it proposes a deep variational EM framework, which to the best of our knowledge has never been proposed before". There are many recent works that use deep variational EM frameworks in different fields, see e.g. this recent review on dynamical variational autoencoders arxiv.org/pdf/2008.12595.pdf which cites a number of them, e.g., https://arxiv.org/pdf/2008.12595.pdf or https://arxiv.org/pdf/1905.01209.pdf .
>
> In general, I found that the authors gave convincing responses to most of the reviewers' remarks. It seems to me that some reviewers gave a low score although their review is relatively short and does not point to a specific major flaw. Overall, I am willing to keep my review and score as they are, since I think this is an interesting, sufficiently novel and technically solid paper.

---

> > ### Author Response · Authors · 2021-08-25
> > **Comparison to Prior Work**
> >
> > Thank you for your helpful comments and for pointing us to these papers.
> >
> > Although we agree that the cited papers use a mixture of EM and deep learning frameworks, we believe our proposed framework has significant differences and advantages. In particular the past approaches either: 1. approximate the E-step posterior using MCMC sampling or 2. approximate the E-step posterior assuming a Gaussian distribution. In contrast, our method allows for non-Gaussian distributions to be estimated in the E-step efficiently using a generative network. This enables efficient estimation in a variety of inverse problems, even if they are poorly described by Gaussian posteriors. The flexible estimation framework also allows us to more easily formulate inverse problems in terms of physically meaningful latent parameters.

---

### Decision · Program_Chairs · 2021-09-27

**Decision:**

Accept (Poster)

**Comment:**

The problem studies inverse problems with a partially unknown forward operator. The idea is to use a general variational Expectation-Maximization framework aided by a normalizing flow generative network. The paper is focused on blind seismic tomographic and shows very good performance. The comparisons to baselines was a bit confusing but we understand that there are limited prior works on this problem and that the proposed method will easily outperform methods that do not use deep generative models.

There was a debate on this paper solving blind inversion versus that mismatch. In any case there is good novelty in the paper which contains novel ideas, is solid in a real and useful application and is very well written.